# Alkaline Stress Induces Different Physiological, Hormonal and Gene Expression Responses in Diploid and Autotetraploid Rice

**DOI:** 10.3390/ijms23105561

**Published:** 2022-05-16

**Authors:** Ningning Wang, Xuhong Fan, Yujie Lin, Zhe Li, Yingkai Wang, Yiming Zhou, Weilong Meng, Zhanwu Peng, Chunying Zhang, Jian Ma

**Affiliations:** 1Faculty of Agronomy, Jilin Agricultural University, Changchun 130000, China; ningningw@jlau.edu.cn (N.W.); linyujie97@163.com (Y.L.); lzm10170212@163.com (Z.L.); wangyingkai99@163.com (Y.W.); zhouyiming2020@163.com (Y.Z.); mengweilong0212@163.com (W.M.); zhangchunying1986@yeah.net (C.Z.); 2Soybean Research Institute, Jilin Academy of Agricultural Sciences, Changchun 130033, China; fxher@126.com; 3Information Center, Jilin Agricultural University, Changchun 130000, China; pengzhanwu@jlau.edu.cn

**Keywords:** diploid, autotetraploid, gene expression, hormone, alkaline stress

## Abstract

Saline−alkaline stress is a critical abiotic stress that negatively affects plants’ growth and development. Considerably higher enhancements in plant tolerance to saline−alkaline stress have often been observed in polyploid plants compared to their diploid relatives, the underlying mechanism of which remains elusive. In this study, we explored the variations in morphological and physiological characteristics, phytohormones, and genome-wide gene expression between an autotetraploid rice and its diploid relative in response to alkaline stress. It was observed that the polyploidization in the autotetraploid rice imparted a higher level of alkaline tolerance than in its diploid relative. An eclectic array of physiological parameters commonly used for abiotic stress, such as proline, soluble sugars, and malondialdehyde, together with the activities of some selected antioxidant enzymes, was analyzed at five time points in the first 24 h following the alkaline stress treatment between the diploid and autotetraploid rice. Phytohormones, such as abscisic acid and indole-3-acetic acid were also comparatively evaluated between the two types of rice with different ploidy levels under alkaline stress. Transcriptomic analysis revealed that gene expression patterns were altered in accordance with the variations in the cellular levels of phytohormones between diploid and autotetraploid plants upon alkaline stress. In particular, the expression of genes related to peroxide and transcription factors was substantially upregulated in autotetraploid plants compared to diploid plants in response to the alkaline stress treatment. In essence, diploid and autotetraploid rice plants exhibited differential gene expression patterns in response to the alkaline stress, which may shed more light on the mechanism underpinning the ameliorated plant tolerance to alkaline stress following genome duplication.

## 1. Introduction

Whole-genome duplication (WGD) or polyploidization plays a vital role in promoting genetic and phenotypic diversities, especially in flowering plants [1,2,3]. It has been documented that 30–70% of plant species have undergone polyploidization or partial genome duplication to some extent during speciation and evolution [4,5,6]. In accordance with the origin and level of WGD, polyploids can be classified into allopolyploids and autopolyploids [6,7,8,9]. In allopolyploids, two distinct genomes are combined through interspecific crosses, whereas autopolyploids are formed by chromosome doubling due to the fusion of unreduced gametes in a species with a lower ploidy level [10,11]. Polyploidization accounts for a series of changes in morphology, physiology, hormonal signal transduction, gene expression, epigenetic modification, and transposon activation of plants as the result of “genome shock”, which is the term coined by McClintock in 1984 to describe the massive genome reorganization and structural changes as the result of chromosome duplication [9,12]. It is generally recognized that polyploidization renders sessile plants more adaptable to extreme environments with a reduced risk of extinction [13]. Conceivably, the duplicated genes in polyploid plants seem to play an imperative role in crop domestication and stress resistance [14,15]. For instance, in rice, which is one of the most important staple food crops worldwide significant phenotypic alterations have been recorded in tetraploids in comparison to diploids, with chromosome doubling occurring in concurrence with a dwindling plant height, broadening and darkening of plant leaves, seed enlargement, an increase in grain weight, and amelioration in environmental adaptability [15,16,17].

Saline−alkaline stress is a common abiotic stress factor that affects plant growth and productivity worldwide [18]. In particular, saline−alkaline stress causes plant metabolic disorders by inducing osmotic pressure and the accumulation of reactive oxygen species (ROS) [19,20,21]. In response to saline−alkaline stress, plants manifest regulation of a series of antioxidant enzymes, such as superoxide dismutase (SOD), catalase (CAT), and peroxidase (POD), and antioxidants that reduce ROS activity, to alleviate oxidative damages and maintain the structural integrity of the cellular membranes by reducing malondialdehyde (MDA) as the result of lipid peroxidation [22,23], and alter the levels of proline and soluble sugars that are essential for osmotic adjustment [23,24].

Although saline stress and alkaline stress normally occur simultaneously, they are in fact regulated by two distinct yet intertwined mechanisms. Saline stress is mainly caused by neutral salts, which is exacerbated by high pH values (>8.5) that severely compromise cell membrane integrity and weaken root vitality and photosynthetic functions [24,25,26]. Accordingly, the severity of salinization can be graded by soil pH values as mild (pH 7.1–8.5), moderate (pH 8.5–9.5), and severe (pH > 9.5), despite the underlying mechanism remaining elusive [27]. The alkaline conditions in soils with elevated pH values primarily cause alkaline stress in plants, which hinders plant growth and development by causing ionic and osmotic stresses that disrupt plants’ physiological and biochemical metabolism [24,28]. In response to alkaline stress, plants manifest substantial changes in cellular phytohormone levels and manifold gene expression patterns in genome-wide gene networks.

Phytohormones exhibit considerable spatiotemporal variations in response to different environmental conditions [29,30,31]. The major phytohormones encompass auxins, gibberellins (GA), cytokinins (CK), abscisic acid (ABA), ethylene (ET), salicylic acid (SA), jasmonates (JA), and brassinosteroids (BR), among which ABA, SA, JA, and ET play particularly critical roles in signaling pathways in response to abiotic stresses [18,32,33].

ABA responds to saline stress by regulating osmotic stress and ROS activity, which are associated with rapid increases in GA, CK, ET, SA, and JA levels during plant adaptation to osmotic stress [34,35]. Saline stress also induces ET accumulation through crosstalks with ABA during salt adaptation in plants [31,36], while JA mediates growth repression and SA enhances the antioxidant system, osmolyte synthesis, and photosynthesis [18,20,37]. In addition, IAA and CK are abundant in root tips to sustain plant growth and development in high-pH growth environments [18,38].

The cellular levels of phytohormones are coordinated by the expression of multiple hormone-related genes [33,39]. For instance, the expressions of *NCED3*, *NCED5*, and *AAO3* were associated with ABA accumulation, while the expressions of JA-related genes, such as *MYC2*, *TAT3*, and *JAZ6,* were induced by dehydration stress in *Arabidopsis thaliana* [18,39]. In sorghum, the expression of ABA-related genes, such as *SbNCED3*, *SbPP2C09*, and *SbPP2C23,* was elevated in response to saline−alkaline stress [40]. Similarly, the expressions of a number of IAA-related genes such as *ARF5*, *GH3.6*, *SAUR36*, *SAUR32* and CK-related genes such as *IPT5* were significantly enhanced concomitant with increases in the levels of IAA and CK in apple rootstocks under alkaline stress [26,41]. Enhancement in salt tolerance was observed in tetraploid rice derived from the japonica rice cultivar *Nipponbare*, as the result of reduced sodium uptake, which was correlated with epigenetic regulation of JA-related genes [15,42]. The commonly cultivated rice is a diploid plant species, but tetraploid rice has been generated for experimental purposes, which is constrained by a poor germination rate and is therefore not suitable for widespread planting [43,44]. Nevertheless, tetraploid rice provides a unique germplasm resource for studying gene dosage and genome evolution, not least with respect to the phenotypic flexibility and adaptability of polyploid plants in response to various environmental stresses. In the advent of deep-sequencing technology and the availability of copious genome databases, it has become feasible to explore the molecular regulation mechanism underpinning the phenotypic variations caused by different ploidy levels in response to stress treatment using a genome-wide approach [44].

In this study, we investigated the phenotypic variations between diploid and autotetraploid rice plants in the early stage of alkaline treatment. A number of key metabolites, key enzymes involved in autooxidation, and phytohormones were analyzed to compare their quantitative responses to autopolyploidization and alkaline stress. Transcriptomic analysis of gene expression was performed to investigate the mechanisms that underpin the variations in response to alkaline stress in diploid and autotetraploid rice plants.

## 2. Results

### 2.1. Effect of Polyploidization on Phenotype

*Oryza sativa* L. ssp. *indica* ‘Yangdao 6’ cultivar 93-11 that is a diploid rice herein referred to as 9311-2x, and its corresponding autotetraploid known as 9311-4x [45] were used in this study. Phenotypic variations in 9311-2x and 9311-4x plants at the three-leaf stage were observed following a period of 6 hours (h) of high-alkaline stress (NaOH, pH 11.39) (Figure 1A). All the alkaline-treated plants showed an overt wilting symptom, which was more marked in 9311-2x plants than in 9311-4x plants. However, there was no significant phenotypic variation between 9311-2x and 9311-4x plants until 7 days (d) following the alkaline treatment (Figure 1B). Plant height, dry weight, and fresh weight of shoot samples were measured and are presented in Appendix A. The dry and fresh weights of plant shoots after the 7 d alkaline treatment were used to calculate the inhibition rate, and they were both higher in 9311-2x plants than in 9311-4x plants (Figure 1C,D). It is therefore apparent that 9311-4x plants exhibited higher tolerance to alkaline stress in comparison to 9311-2x plants.

### 2.2. Effect of Polyploidization on Physiological Parameters

To investigate the biochemical responses to the alkaline treatment, the contents of proline, soluble sugars, and MDA, as well as SOD and POD enzyme activities in 9311-2x and 9311-4x plants were measured at five successive time points (0, 3, 6, 12, and 24 h) following the high alkaline treatment (Figure 2). Proline contents in 9311-2x and 9311-4x plants were decreased by 2.59% and 7.21% at 24 h, respectively. The soluble sugar contents were increased by 7.44% and 9.07% at 3 h before declining by 5.15% and 4.32% at 24 h in 9311-2x and 9311-4x plants, respectively. SOD activity showed a sustained increase throughout the alkaline treatment in 9311-4x plants, which was in contrast to the delayed response of increase at 24 h in 9311-2x plants. POD activity showed an initial elevation in the early stage of the alkaline treatment but substantially reduced thereafter in both 9311-2x and 9311-4x plants. The MDA content was increased in both 9311-2x and 9311-4x plants in response to the alkaline treatment, with the highest level being recorded at 6 h after treatment in 9311-4x. Taken together, it appears that these selected physiological parameters that are commonly used for abiotic stress were variable at the five time points in the first 24 h following alkaline stress treatment.

### 2.3. Effect of Polyploidization on Hormone Levels

The contents of a number of phytohormones in response to the alkaline treatment were comparatively analyzed between diploid and tetraploid rice plants (Figure 3). Rapid responses of an increase in ABA levels were observed in both 9311-2x and 9311-4x plants following the alkaline treatment, with significantly higher levels in 9311-4x at 3 h and 24 h than in 9311-2x at the same time points, while the opposite was true at both 6 h and 12 h after treatment. This is well in line with previous studies which found that ABA is essential for osmotic stress and plays an important role in response to high pH stress in rice [46]. Similarly, JA also exhibited a rapid and dramatic response by increasing its content in the first 12 h following the alkaline treatment, with a significantly higher response in 9311-4x than in 9311-2x at 3 h, congruent with previous findings [39]. Saline stress has been shown to inhibit root length by affecting JA activity, which is associated with ABA signal transduction under saline stress conditions [18]. Distinct from the rapid and dramatic response of ABA and JA, the content of IAA showed a rather moderate or ambiguous response to the alkaline treatment in both diploid and tetraploid rice plants, which is consistent with previous findings in Arabidopsis after dehydration stress [39]. It also appears that IAA levels in 9311-4x at various sampling time points fluctuated lesser than in 9311-2x plants. The smallest variations were observed in CK (trans-zeatin, tZ) levels, with very moderate yet significant increases in 9311-4x plants in response to the alkaline treatment, somewhat similar to previous findings in Arabidopsis where tZ levels were initially increased at 24 h, before returning to control levels at 48 h, and significantly reduced at 72 h [39]. Under alkaline stress, the ACC contents were drastically reduced in 9311-2x but remained constant in 9311-4x. The SA contents showed a rather moderate reduction in 9311-2x, but a moderate increase in 9311-4x plants in response to the alkaline treatment.

### 2.4. Effect of Polyploidization on Gene Expression

To explore the mechanism underpinning the higher alkaline tolerance in autotetraploid rice than in diploid rice, we analyzed and compared the genome-wide gene expression profiles in roots and shoots derived from 9311-2x and 9311-4x plants after 6 h of alkaline treatment. R2 and R4 represent the root samples derived from 9311-2x and 9311-4x plants, respectively. R2_AS and R4_AS represent the alkaline stress treated root samples derived from 9311-2x and 9311-4x plants, respectively. Likewise, Sh2 and Sh4 represent the shoot samples derived from 9311-2x and 9311-4x plants, respectively. Sh2_AS and Sh4_AS represent the alkaline stress treated shoot samples derived from 9311-2x and 9311-4x plants, respectively. Transcriptomic analysis detected 29,481, 30,354, 29,468, 30,107, 30,721, 30,607, 30,474, and 30,502 genes in the R2, R2_AS, R4, R4_AS, Sh2, Sh2_AS, Sh4, and Sh4_AS samples, respectively. The quality of the clean reads is outlined in Appendix A. As shown in Figure 4A, copious genes were differentially expressed in response to alkaline stress. The co-expressed and specifically-expressed genes in diploid and tetraploid rice plants can be classified into three groups using Venn diagram analysis (Figure 4B,C). In the roots of polyploid plants, 3.2% of genes were specifically expressed in R4, compared to 2.4% in R2, 2.7% in R4_AS, and 3.5% in R2_AS. In the shoots, 3.2% of genes were specifically expressed in Sh4, compared to 7.7% in Sh2, 3.6% in Sh4_AS, and 3.2% in Sh2_AS. Under alkaline stress, 4.1% of genes were specifically expressed in R2_AS, 3.5% in R2, and 3.6% in R4_AS, while 4.6% of genes were specifically expressed in R4 roots. As for spatial gene regulation, 12.1% of genes were specifically expressed in Sh2, compared to 7.6% in R2, 9.8% in Sh4, and 10.4% in R4. Similarly, 11.2% of genes were specifically expressed in Sh2_AS compared to 8.8% in R2_AS, 12.1% in Sh4_AS, and 8.7% in R4_AS. There exist copious specifically expressed genes that were induced by alkaline stress, polyploidization, and tissue specificity, among which the spatial-specific genes were the most abundant.

### 2.5. Differentially Expressed Genes (DEGs) between Diploid and Autotetraploid Rice under Alkaline Treatment

DEGs between diploid and autotetraploid rice plants under alkaline stress were profiled by performing whole-genome gene expression and transcriptomic analyses. A total of 12 DEG groups were identified between 9311-2x and 9311-4x plants under alkaline stress, concerning tissue specificity or ploidy level (Figure 5A). Under alkaline stress, there were more DEGs between Sh4_AS and Sh4 shoots (2733 DEGs: 2063 upregulated, 670 downregulated) than between Sh2_AS and Sh2 shoots (1692 DEGs: 910 upregulated, 782 downregulated), and a higher number of DEGs between R4_AS and R4 roots (1656 DEGs: 1046 upregulated, 610 downregulated) than between R2_AS and R2 roots (1534 DEGs: 1032 upregulated, 502 downregulated). It is therefore conceivable that there are more DEGs in 9311-4x plants relative to 9311-2x plants in response to alkaline stress. In terms of tissue specificity, 7358 DEGs (4346 upregulated, 3013 downregulated) were identified between Sh2 and R2, compared to 7209 DEGs (3783 upregulated, 3426 downregulated) between Sh4 and R4. Moreover, 7275 DEGs (4032 upregulated, 3243 downregulated) and 7760 DEGs (4457 upregulated, 3303 downregulated) were identified for Sh2_AS vs. R2_AS and Sh4_AS vs. R4_AS, respectively. These findings suggest that there were many tissue-specific DEGs in the 9311-2x and 9311-4x plants and that the number of DEGs increased in response to alkaline stress. As for the ploidy level, 3380 (1275 upregulated, 2105 downregulated), 354 (249 upregulated, 105 downregulated), 1328 (737 upregulated, 591 downregulated), and 643 (324 upregulated and 319 downregulated) DEGs were found in Sh4 vs. Sh2, R4 vs. R2, Sh4_AS vs. Sh2_AS, and R4_AS vs. R2_AS, respectively.

Venn analyses were performed to compare the co-expressed and specifically-expressed DEGs between groups (Figure 5B,C). A total of 596 specific DEGs were identified in 9311-2x (R2_AS vs. R2) compared to 691 in 9311-4x (R4_AS vs. R4), suggesting that the mechanism responding to alkaline stress differs between 9311-2x and 9311-4x plants, as confirmed by comparing Sh2_AS vs. Sh2 and Sh4_AS vs. Sh4. In addition, there were more DEGs induced by alkaline stress in shoots than in roots for both 9311-2x plants (1393 for Sh2_AS vs. Sh2; 1235 R2_AS vs. R2) and 9311-4x plants (2449 for Sh4_AS vs. Sh4; 1327 for R4_AS vs. R4), indicating that the gene expression response to alkaline stress differs between shoots and roots. Venn analysis also revealed 1876 DEGs that were specifically expressed in the shoots and roots of 9311-2x plants under mock conditions (Sh2 vs. R2), compared to 1792 under alkaline stress (Sh2_AS vs. R2_AS). In 9311-4x plants, there were 2305 and 2846 specific DEGs in the shoots and roots (Sh4 vs. R4 and Sh4_AS vs. R4_AS, respectively). Conversely, there were 2270 and 2120 specific DEGs between the shoots and roots of 9311-2x and 9311-4x plants (Sh2 vs. R2 and Sh4 vs. R4, respectively), compared to 1578 and 2063 DEGs under alkaline stress (Sh2_AS vs. R2_AS and Sh4_AS vs. R4_AS, respectively). Thus, gene expression appears to display tissue specificity between the shoots and roots of rice and is induced by alkaline stress, particularly in 9311-4x plants. Finally, we compared the DEGs between the shoots and roots of 9311-4x and 9311-2x plants (Sh4 vs. Sh2 and R4 vs. R2/ Sh4_AS vs. Sh2_AS and R4_AS vs. R2_AS) and the DEGs in 9311-4x and 9311-2x plants under mock conditions compared to alkaline stress conditions in roots and shoots (R4 vs. R2 and R4_AS vs. R2_AS/ Sh4 vs. Sh2 and Sh4_AS *vs* Sh2_AS). These results confirm that gene expression differs between diploid and autotetraploid rice plants under alkaline stress, to a greater extent in autotetraploid than in diploid plants.

### 2.6. Gene Ontology (GO) and Kyoto Encyclopedia of Genes and Genomes (KEGG) Enrichment Analyses of DEGs between Diploid and Autotetraploid Rice

To examine the function of the DEGs identified between the different groups, we performed both GO (Figure 6A and Appendix A) and KEGG (Figure 6B and Appendix A) enrichment analyses. For the Sh2_AS vs. Sh2 DEGs, the most enriched GO terms for Biological Process (BP) were lipid transport (GO:0006869), apoptotic process (GO:0006915), lipid localization (GO:0010876), and diterpenoid metabolic process (GO:0016101); those for Cellular Component (CC) were membrane-bound organelles (GO:0043227) and intracellular membrane-bounded organelle (GO:0043231); and those for Molecular Function (MF) were electron carrier activity (GO:0009055), monooxygenase activity (GO:0004497), and glutathione transferase activity (GO:0004364). For the Sh4_AS vs. Sh4 DEGs, the most enriched GO terms for BP were apoptotic process (GO:0006915), microtubule-based movement (GO:0007018), and movement of cell or subcellular component (GO:0006928); those for CC were cell part (GO:0044464) and cell (GO:0005623); and those for MF were electron carrier activity (GO:0009055), receptor activity (GO:0004872), and protein tyrosine kinase activity (GO:0004713). For the Sh4 vs. Sh2 DEGs, the most enriched GO terms for BP were apoptotic process (GO:0006915), DNA replication (GO:0006260), and protein phosphorylation (GO:0006468); those for CC were cell part (GO:0044464) and cell (GO:0005623); and those for MF were electron carrier activity (GO:0009055), receptor activity (GO:0004872), and molecular transducer activity (GO:0060089). For the Sh4_AS vs. Sh2_AS DEGs, the most enriched GO terms for BP were isoprene metabolic process (GO:0006720), lignin catabolic process (GO:0046274), and phenylpropanoid catabolic process (GO:0046271); those for CC were membrane-bound organelles (GO:0043227) and intracellular membrane-bounded organelle (GO:0043231); and those for MF were electron carrier activity (GO:0009055), monooxygenase activity (GO:0004497), and oxidoreductase activity (GO:0016491). The corresponding GO enrichment analysis for rice root samples is shown in Appendix A.

According to the KEGG enrichment analysis, under alkaline stress, the biosynthesis of secondary metabolites, metabolic pathways, cutin, suberine, and wax biosynthesis, and phenylpropanoid biosynthesis were significantly enriched in both Sh2 and Sh4. The corresponding KEGG enrichment analysis of the rice root samples is shown in Appendix A. Interestingly, the most enriched MF terms from the GO analysis were related to oxidoreductase activity, glutathione transferase activity, and ATP binding (Figure 6C, Appendix A). When the DEGs were mapped onto the annotated sequence (*Osativa_323_v7.0.annotation_info*), DEGs belonging to the peroxidase protein superfamily were also detected, most of which were significantly upregulated in Sh4_AS relative to Sh4. It is conceivable that a higher peroxidase expression may convey higher tolerance to alkaline stress in autotetraploid rice than diploid progenitors.

### 2.7. Phytohormone Related DEGs Vary with Ploidy Level

To investigate the DEGs involved in hormone metabolism under alkaline stress between diploid and autotetraploid rice, MapMan software was used to analyze the genes that are associated with phytohormones. A total of 130 DEGs were identified that are involved in the metabolism of ABA, ET, CK, SA, JA, and GA (Figure 7, Appendix A). For example, the expression of the 9-cisepoxycarotenoid dioxygenase 3 (*NCED3*) gene was upregulated in roots after alkaline stress and *NCED4* was upregulated in shoots in response to alkaline stress, particularly in autotetraploid rice. The involvement of these genes in ABA biosynthesis has been well documented [39]. The upregulation of *NCED* expression is in line with the increase in ABA levels in shoots following the alkaline treatment for 6 h. *NCED4* gene expression was significantly elevated in Sh4 vs. Sh2 and Sh4_AS vs. Sh2_AS, and SA levels in shoots of 9311-4x rice were lower than in 9311-2x at 6 h.

In Arabidopsis, it has been reported that the *Lipoxygenase2* (*LOX2)* and *Allene oxide synthase* (*AOS*) genes are upregulated in response to saline stress and are associated with elevated JA levels [39]. Here, in rice, *LOX2* homologs were upregulated after saline stress. Two *AOS* genes were identified in Sh2_AS, R2_AS, Sh4_AS, and R4_AS in abundance, being upregulated by > 2-fold in 9311-4x relative to 9311-2x in both root and shoot tissues. Further, the *LOX2* and *AOS* genes were also upregulated by alkaline stress at 6 h in both 9311-2x and 9311-4x plants.

The *GRETCHEN HAGEN3 (GH3)* gene family maintains auxin homeostasis by conjugating excess IAA to amino acids [47,48,49]. *GH3.6* was significantly downregulated in Sh4 vs. Sh2 but upregulated in Sh4_AS vs. Sh4 and Sh4_AS vs. Sh2_AS. In light of the gene expression data, IAA-related genes appeared to be downregulated by polyploidization and upregulated by alkaline stress; however, there was no significant alterations in IAA levels at 6 h in the shoot samples. Adenosine phosphate-isopentenyl transferase *(IPT) 3,* a key enzyme in tZ metabolism, was significantly downregulated in Sh4_AS vs. Sh4 and upregulated in Sh4 vs. Sh2. In addition, *CK oxidase* (*CKX*) genes, including *CKX1*, *CKX3*, *CKX4*, *CKX5*, and *CKX6* which are involved in CK degradation were induced by alkaline stress. In particular, *CKX1* gene expression was upregulated by > 2-fold in R2_AS vs. R2 and R4 vs. R2, consistent with the alterations in tZ hormone levels between diploid and autotetraploid plants after alkaline stress for 6 h.

*ACC synthase* (*ACS*) genes are known to play key roles in ethylene biosynthesis [18]. *ACS6* was preferentially expressed in roots and shoots following alkaline stress, with exceptionally high expression in R4_AS. *ERF11* has been reported to negatively regulate ET biosynthesis by directly repressing *ACSh2* and *ACS5* in response to ABA [50]. Notably, the *ERF1*, *ERF9*, and *ERF12* genes were upregulated by the alkaline treatment in both the shoots and roots of diploid rice plants, and > 2-fold higher in autotetraploid compared to diploid rice plants. In addition, the upregulation of *ACS* genes was associated with ACC levels, which declined in the shoots of diploid plants but remained unchanged in autotetraploid after 6 h of alkaline stress, likely due to the unregulated expression of ERF-related genes. Two *SA methyltransferase* (*BSMT1*) gene was significantly upregulated between R4_AS vs. R4, Sh4_AS vs. Sh4, R4_AS vs. R2_AS, and Sh4_AS vs. Sh2_AS. These observations are well in line with the upregulation of *BSMT1* by dehydration stress in Arabidopsis [39].

## 3. Discussion

The rice cultivar 93-11 has been reported to undergo genome-wide changes in gene expression by saline−alkaline stress [51,52,53]. In addition, various epigenetic markers have been correlated with the expression patterns of the accessible chromatin region (ACR) gene in 9311-2x and 9311-4x plants, suggesting that tetraploidization alters rice plant morphology and products by modulating chromatin signatures and transcriptional profiling [53].

In the early stage of saline−alkali stress, the accumulation of sodium ions (Na^+^) induces instantaneous osmotic pressure on plant cells [26,54]. In addition, plants exposed to moderate and severe alkaline stress conditions accumulate relatively high levels of carbohydrates and display significantly increased ROS and MDA contents [24,55]. To adapt to osmotic stress and physiological drought, plants accumulate various molecular compounds and increase oxidase activity [39,56]. MDA and ROS are generated by membrane lipid peroxidation in response to abiotic stress in plants, and their accumulation can aggravate cellular damage. However, the antioxidant enzymes SOD and POD can scavenge ROS in response to abiotic stress, whereby SOD reduces O^2−^ to H_2_O_2_, and POD converts H_2_O_2_ into water and oxygen. We found that MDA, SOD, and POD contents differed between the 9311-4x and 9311-2x plants in the early stages after alkaline stress, indicating that the mechanism of plant resistance to alkaline stress differs between autotetraploid and diploid rice plants.

To further elucidate the mechanism underpinning the response of rice to alkaline stress, we performed transcriptome analysis, which confirmed that the gene expression induced by alkaline stress varied significantly between diploid and autotetraploid plants. In this study, tissue specificity exerted a greater effect on gene expression than alkaline stress; nevertheless, the plant responses to polyploidization and alkaline stress did impact the expression of a plethora of genes across the rice genome. Subsequent GO enrichment analyses revealed that the expression of genes related to apoptosis, electron carrier activity, and monooxygenase activity was significantly enriched under alkaline stress, indicating that the differential expression of these genes may be required in response to alkaline stress in rice. KEGG analysis further suggested that secondary metabolites and phenylpropanoid biosynthesis play essential roles in the resistance of rice plants to alkaline stress. In particular, the pathways of cutin, suberine, and wax biosynthesis were significantly enriched in rice shoots, whereas starch and sucrose metabolisms were significantly enriched in the roots, indicating distinct yet connected mechanisms that orchestrate the tissue specific responses to alkaline stress. It is worth noting that diterpenoid-related pathways were significantly enriched in the shoots of diploid rice and the roots of autotetraploid rice as demonstrated by both GO and KEGG analyses, underpinning the enhanced tolerance against alkaline stress through polyploidization. GO analysis revealed the enrichment of genes related to glutathione transferase activity, oxidoreductase activity, and ATP binding, which are all involved in oxidation. Overall, the DEGs related to peroxidase superfamily proteins were significantly upregulated by alkaline stress in autotetraploid rice, which is likely attributable to the observed higher resilience against stress in autotetraploid relative to diploid rice.

Saline−alkaline stress causes osmotic and ionic stresses in plant cells due to dehydration; however, specific ions (Na^+^, K^+^) are upregulated under dehydration stress to maintain cellular homeostasis. Excess Na^+^ uptake has been shown to affect the NH_4_^+^ assimilation pathway while inhibiting the glutamate synthase pathway and promoting the glutamate dehydrogenase pathway, thereby resulting in leaf senescence [57]. Alkaline stress induced the expression of an eclectic array of genes involved in ion transport in 9311-2x and 9311-4x plants (Appendix A). Notably, the *OsGS1;3* (*Os03g0712800*) gene, involved in nitrogen metabolism, was significantly upregulated in both autotetraploid and diploid rice strains after alkaline stress. *OsAMT1;1* (*Os04g0509600*) and *OsHKT1;1* (*Os04g0607500*) play key roles in NH_4_^+^ and Na^+^ transport, respectively, which were significantly upregulated in diploid and autotetraploid plants under mock and alkaline stress conditions. Furthermore, the *OsHAK1* gene (*Os04g0401700*), which is involved in K^+^ transport, was significantly upregulated by > 4-fold as the result of chromosome doubling in 9311-4x under the mock conditions but was expressed at similar levels in 9311-2x plants under alkaline stress.

Phytohormones regulate plant growth and stress responses. Previous studies have reported that saline stress can seriously affect plant growth and development by altering stress and growth hormones, whereas plants can mediate salt tolerance by increasing ABA, JA, SA, SL, CK, and ET while decreasing GA, BR, and auxin [18]. In this study, we found that the levels of phytohormones including ABA, JA, tZ, ACC, and SA showed considerable fluctuations between 9311-2x and 9311-4x in the first 24 h following alkaline stress; however, most of them became consistently more abundant in 9311-4x than in 9311-2x after alkaline stress for 24 h, reflecting the stabilized hormonal changes in response to alkaline stress over time [39]. Taken together, these observations indicated that the enhancements in cellular phytohormone levels could have contributed to the amelioration in alkaline tolerance in autotetraploid rice.

It has been well documented that the accumulations of proline and soluble sugars in plant roots augment in response to exogenous ABA, thereby reducing osmotic pressure and increasing water retention to improve root water uptake and transport [33]. In addition, JA was found to reduce the plant height and root growth and suppress seed germination and RuBP carboxylase biosynthesis while promoting leaf senescence, stomatal closure, and the synthesis of leaf protease inhibitors [33]. It is therefore conceivable that the elevation in JA accumulation fostered by alkaline stress may curtail plant cellular catabolism and stomatal opening to reduce transpiration and maintain water homeostasis. The negative association between ABA and JA levels observed in the first 24 h following the alkaline treatment is in agreement with previous studies by Urano et al [39].

The mechanisms that govern plant tolerance to abiotic stress depend on phytohormones both singly and in combination. Previous studies reported that abiotic stresses regulate the expression of IAA responsive genes and IAA synthesis [33], and that IAA and GA act in concert to promote polyamine production induced by saline stress [58]. ABA, on the other hand, is a negative regulator of polyamine synthesis, playing a crucial role in maintaining homeostasis of competitive syntheses of polyamine and ACC [59]. IAA mediates the ACC reaction, which in turn promotes IAA synthesis [18]. In this study, the accumulation of IAA was elevated in the diploid rice but inhibited in the autotetraploid rice at 6-12 h after alkaline stress, whereas ACC was downregulated in the diploid rice and maintained relatively constant in the autotetraploid rice. 

SA inhibits peroxidase activity to enhance H_2_O_2_ and O^2−^ accumulation while activating SOD and other enzymes to increase plant tolerance to various abiotic stresses [26]. Further, SA can also regulate the levels of free amino acids and soluble proteins to reduce cell membrane lipid peroxidation and maintain cellular metabolic activity. SOD, POD, and catalase activities can increase the levels of antioxidants, glutathione, and ascorbic acid to scavenge and eliminate ROS, alleviate cellular damage, and protect the cell membrane structure to maintain ionic and osmotic stabilities [26]. In 9311-4x rice plants, SOD activity was slightly increased in concurrence with the increase in SA accumulation at 6–12 h following alkaline stress, validating its functional role in imparting a higher level of alkaline tolerance in autotetraploid rice than diploid rice. Furthermore, the tZ level was significantly elevated at 12–24 h following the alkaline treatment, more prominently in 9311-4x than in 9311-2x plants, which may suggest that tZ plays an important role in plant responses to alkaline stress in the early stage of the stress treatment [39]. Together with previous studies, our results support the notion that cellular levels of phytohormones are controlled by the coordinated expressions of a suite of genes that are modulated by abiotic stresses [33,39,42]. Nevertheless, it is conceivable that the relationship between gene expression and phytohormone accumulation is complex, especially in autotetraploid plants where the duplicated genes may have gained refined or new roles through functional diversification. It is tempting to assume that the sophisticated responses to alkaline stress in allotetraploid rice could be conducive to the normal plant development without fitness trade-offs. Furthermore, the key genes and genetic networks related to the alterations in cellular phytohormone levels in polyploid plants in response to alkaline stress are intriguing and warrant further investigation.

The genome-wide analysis of rice transcription factors led to the identification of 1594 genes from 56 families induced by polyploidization and alkaline stress, among which AP2, bHLH, bZIP, C2H2, Dof, ERF, HD-ZIP, HSF, LBD, MYB, MYB-related, NAC, WRKY, and those genes related to phytohormones were the most prominent (Appendix A).

Overall, the findings embodied in this study lend further credence to the idea that multiple phytohormones act in concert in response to alkaline stress treatment, which are modulated by gene expression genome-wide. Furthermore, we demonstrated that diploid and autotetraploid rice plants were endowed with different phenotypes, physiological characteristics, phytohormone levels, and gene expression patterns in response to alkaline stress. Relative to the diploid rice, autotetraploid rice plants exhibited greater tolerance to alkaline stress, suggesting that chromosome doubling may render the autotetraploid plants advantageous in conferring stress tolerance, which may shed more light on the evolution of sessile plants in coping with environmental stresses, and facilitate a theoretical chassis for further investigations on genome-wide molecular responses to environmental stresses in polyploid plants.

## 4. Materials and Methods

### 4.1. Plant Materials and Stress Treatments

In contrast to 9311-2x that encompasses 24 chromosomes, 9311-4x is endowed with 48 chromosomes (Appendix A). Root-tip cells were used for karyotypic analysis, whereby chromosomes were stained with 4,6-diamidino-2-phenylindole (DAPI). All the plants used in the experiments had been propagated by selfing for four generations in our laboratory to ensure their stable heredity. The 9311-2x plants were generally taller than the 9311-4x plants; however, 9311-4x had broader and thicker shoots at the grain filling stage and produced larger seeds (Appendix A). Seeds were germinated on moist tissue towels at 28 °C in the dark for 2–3 d. Seedlings were transferred to a half-strength Hoagland nutrient solution to saturate for 20 d, with continuous fluorescent lighting under a temperature cycle of 28/26 °C for 16/8 h. In brief, rice seeds were sowed in a plastic container equipped with hydroponic culture apparatus (60–70% humidity). At the three-leaf stage, alkaline stress was applied for 1 or 7 d by supplementing the nutrient solution that was adjusted to pH 11.39 with NaOH. Fifteen plants with consistent growth were chosen for treatment in each replicate. The fresh weights and dry weights of plants were recorded and the inhibition rates were calculated from three biological replicates, as previously described [42,60].

### 4.2. Measurements of Small-Molecule Organic Compounds, Enzyme Activity, and Hormone Levels

Proline, soluble sugars, MDA levels, and the activity of SOD and POD were detected in the shoots of 9311-2x and 9311-4x plants at 0, 3, 6, 12, and 24 h after alkaline stress. The time point of 0 h was used as a control. The content of proline was measured using the acid ninhydrin method as described by Bates et al [61]. Absorbance was measured at 520 nm using the SmartSpecTM Plus spectrophotometer (BioRad, Hercules, CA, USA). The contents of soluble sugars were measured as previously described [62,63]. In brief, 5 mL anthrone reagent was added to 1 mL extract, and incubated at 95 °C for 15 min. After being cooled to room temperature, absorbance was measured at 625 nm using the aforementioned spectrophotometer (BioRad). The content of MDA was determined following the protocols described by Song *et al.* (2011) [63]. The rice extracts were mixed with 5 mL 10% TCA solution, centrifuged at 10,000 g for 10 min, prior to adding 10% trichloroacetic acid and 0.6% thiobarbituric acid, and incubated in a waterbath at 95 °C for 30 min, stopped with ice. The absorbance of the solution was then measured at 450, 532, and 600 nm. The measurements of SOD and POD activities were conducted as previously described by Yang *et al.* (2012) [62]. Briefly, the 500 mg trituration samples were mixed with 50 mM potassium phosphate buffer (pH 7.8) containing 1% polyvinylpyrrolidone, prior to being centrifuged at 15,000 g for 20 min at 4 °C. The supernatants thus obtained were used for SOD and POD detection, and the absorbance of the solution was then measured at 560nm and 470 nm, respectively. Fifteen plants were pooled together as one sample in each replicate. All samples were tested in three independent experiments with three replicates each.

The contents of phytohormones at the same time points were assayed using an ACQUITY UHPLC system (Waters Corporation, Milford, MA, USA) coupled with an AB SCIEX API 5500 System (AB SCIEX, Framingham, MA, USA). The profile analysis was conducted using the Analyst 1.6.2 workstation (AB SCIEX) at Shanghai Lu-Ming Biotech Co., Ltd. All the phytohormone assays contained three biological replicates, each of which contained three technical replicates.

### 4.3. RNA Isolation and Sequencing Analysis

Total RNA was isolated from the shoots and roots of 9311-2x and 9311-4x plants after 6 h of mock conditions or alkaline stress. A total of eight samples were subjected to transcriptomic analysis, including 9311-2x-shoot, 9311-4x-shoot, 9311-2x-root, 9311-4x-root under mock conditions (Sh2, Sh4, R2, and R4, respectively), or under alkaline stress conditions (Sh2_AS, Sh4_AS, R2_AS, and R4_AS, respectively).

RNA sequencing (RNA-seq) was performed using an Illumina preparation kit (Illumina, San Diego, CA, USA), and the Novaseq platform (125 bp paired-end reads, Q20 > 95%, Q30 > 88%) [60]. Approximately 30-40 million raw sequence reads were obtained for each sample. Adaptors, low-quality sequences, and poly-A sequences were removed to generate clean data for analysis using BWA and STAR, as previously described [64,65]. The reads were mapped to a reference genome using pan-genome [66] and then normalized using the DESeq2 R package to search for DEGs with a max_readcount > 30, foldchange > 2, and *p* < 0.05. A total of 24 libraries consisting of three biological replicates of the eight sample types were analyzed. GO and KEGG enrichment analyses were performed as previously reported [60,67].

### 4.4. Quantitative Real-Time Polymerase Chain Reaction

To validate the RNA-seq data, ten DEGs were randomly selected and analyzed by quantitative real-time polymerase chain reaction (qRT-PCR). RNAs from all the aforementioned samples were extracted and the first strand cDNAs were synthesized by using gene-specific primers (Appendix A). qRT-PCR was conducted using a Top Green qPCR SuperMix kit (TransGen Biotech, Beijing, China), as per the manufacturer’s instructions on a Roche PCR system (manufacturer’s information). The primers were downloaded from https://biodb.swu.edu.cn/qprimerdb/ (accessed on 12 April 2022). Each gene was repeatedly analyzed three times and normalized to the reference gene GADPH (Appendix A) [68].

### 4.5. Statistical Analysis

Using DPS software [69], all statistical analyses were performed with Duncan’s multiple range test or *t*-tests. Data were presented as the mean ± standard error of three biological replicates. The threshold for statistical significance was set at *p* < 0.05.

## Figures and Tables

**Figure 1 ijms-23-05561-f001:**
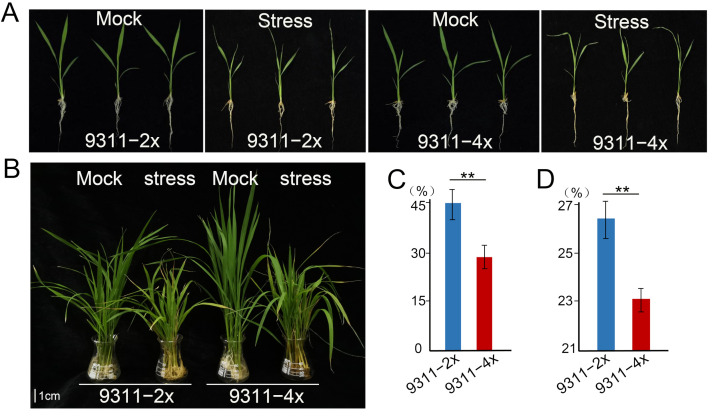
Phenotypic variations in response to alkaline stress in 9311-2x and 9311-4x rice plants. (**A**) Response to alkaline stress for 6 hours (h). (**B**) Response to alkaline stress for 7 days (d). (**C**) Inhibition rate of the dry weight in 9311-2x and 9311-4x following alkaline stress for 7 d. ** *p* < 0.05. (**D**) Inhibition rate of the fresh weight in 9311-2x and 9311-4x following alkaline stress for 7 d. ** *p* < 0.05.

**Figure 2 ijms-23-05561-f002:**
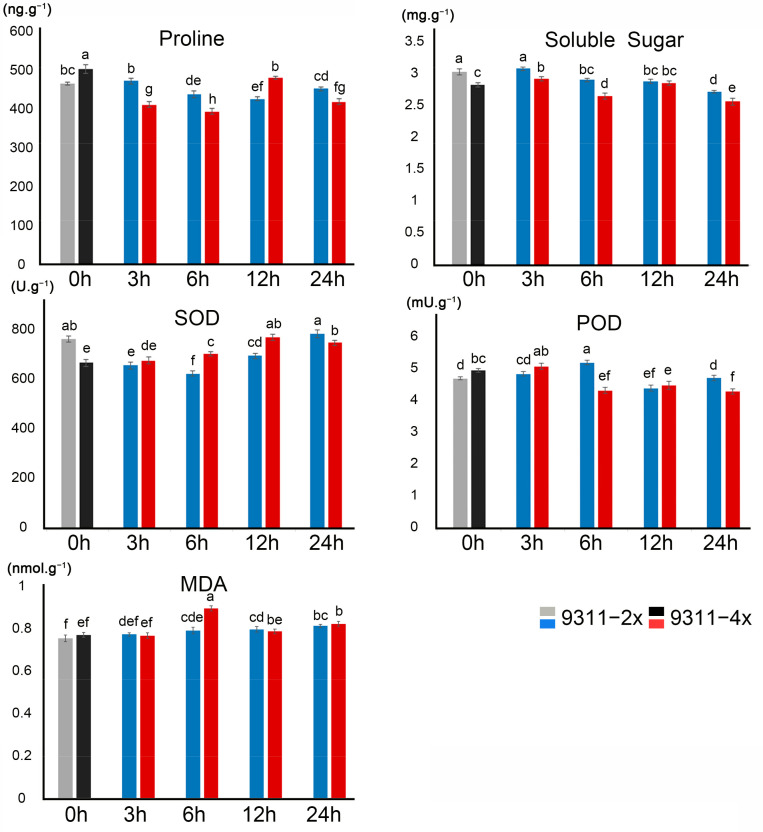
Physiological characteristics between 9311-2x and 9311-4x rice plants under alkaline stress. Proline, soluble sugars, SOD, POD, and MDA were detected at different time points after alkaline stress, including 0 h (without stress), 3 h, 6 h, 12 h, and 24 h. The values marked with different letters between two columns are significantly different at *p* < 0.05.

**Figure 3 ijms-23-05561-f003:**
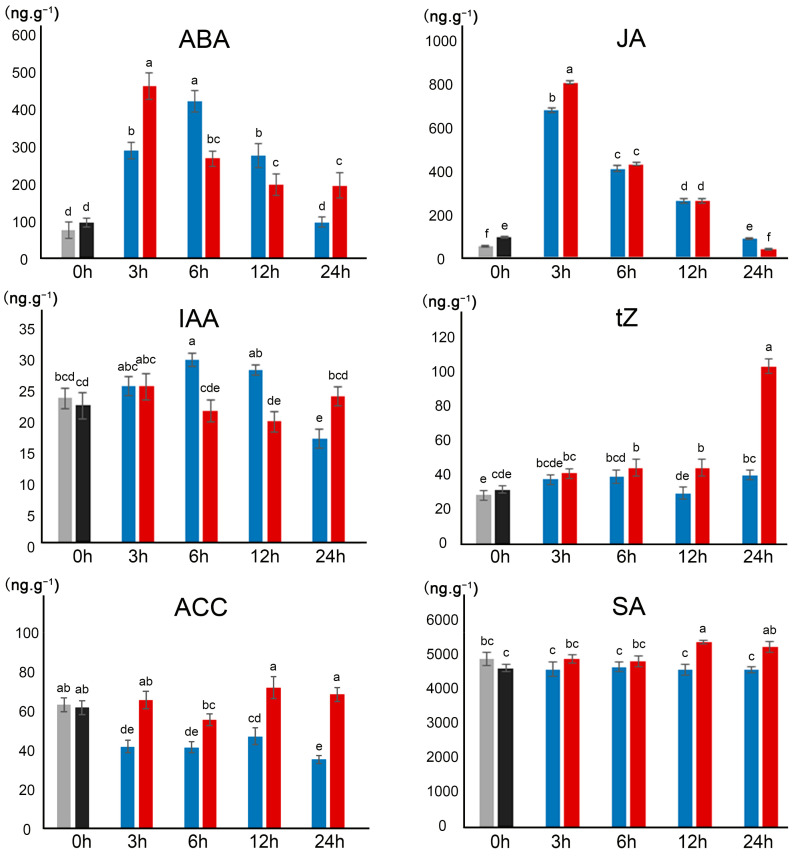
Plant hormones levels between 9311-2x and 9311-4x rice plants under alkaline stress. ABA, JA, IAA, tZ, ACC, and SA were detected at different time points: 0 h (without stress), 3 h, 6 h, 12 h, and 24 h after alkaline stress. The values marked with different letters between two columns are significantly different at *p* < 0.05.

**Figure 4 ijms-23-05561-f004:**
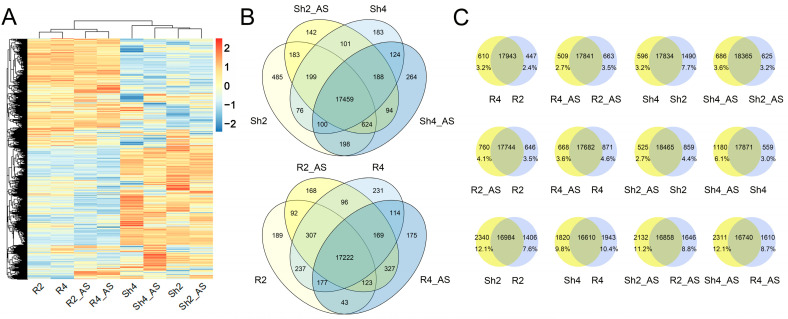
Genome-wide gene expression of roots and shoots between 9311-2x and 9311-4x rice plants under mock and high pH conditions. (**A**) Heatmap of gene expression levels in roots and shoots. R2 and R4 represent the root samples derived from 9311-2x and 9311-4x plants, respectively. R2_AS and R4_AS represent the alkaline stress treated root samples derived from 9311-2x and 9311-4x plants, respectively. Likewise, Sh2 and Sh4 represent the shoot samples derived from 9311-2x and 9311-4x plants, respectively. Sh2_AS and Sh4_AS represent the alkaline stress treated shoot samples derived from 9311-2x and 9311-4x plants, respectively. (**B**) Venn diagram of common and unique gene expression in roots and shoots. (**C**) Details of the gene expression between roots and shoots. Differentially expressed genes between roots and shoots. The number is marked on each column.

**Figure 5 ijms-23-05561-f005:**
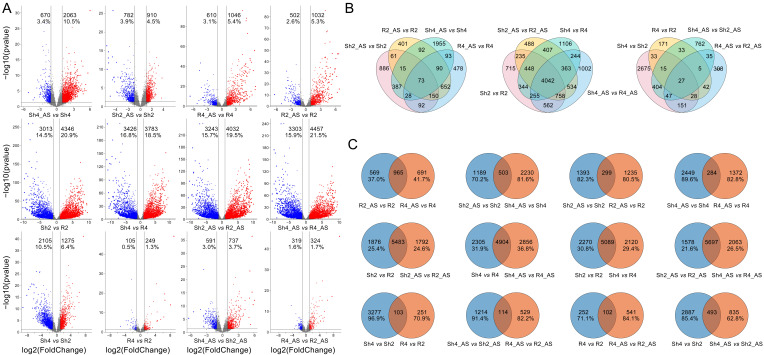
Differentially expressed genes (DEGs) between the roots and shoots of 9311-2x and 9311-4x rice plants under mock and high pH conditions. (**A**) The number of upregulated (blue) and downregulated (red) DEGs, which are marked at the top. (**B**) Common and unique DEGs between roots and shoots. (**C**) Details of the DEGs between all samples. R2 and R4 represent the root samples derived from 9311-2x and 9311-4x plants, respectively. R2_AS and R4_AS represent the alkaline stress treated root samples derived from 9311-2x and 9311-4x plants, respectively. Likewise, Sh2 and Sh4 represent the shoot samples derived from 9311-2x and 9311-4x plants, respectively. Sh2_AS and Sh4_AS represent the alkaline stress treated shoot samples derived from 9311-2x and 9311-4x plants, respectively.

**Figure 6 ijms-23-05561-f006:**
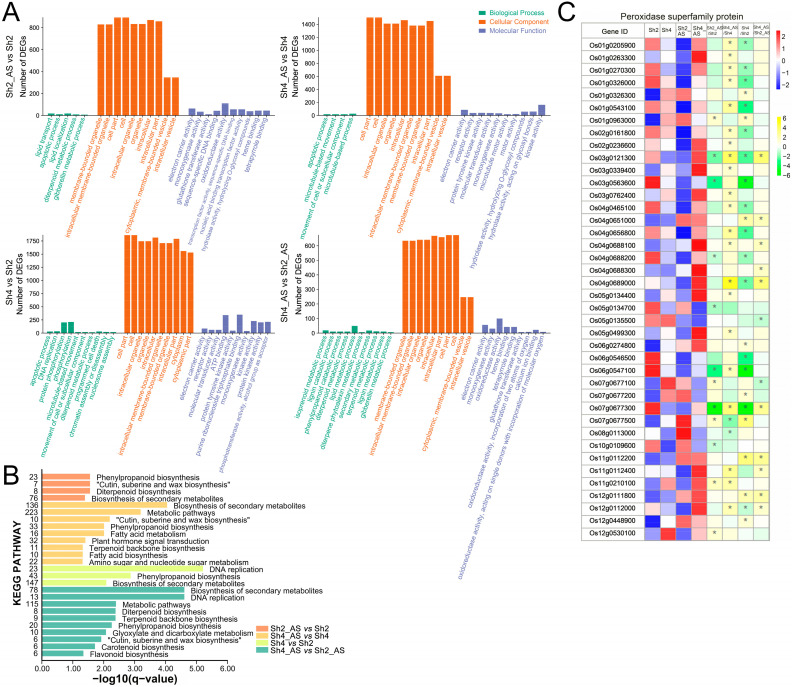
(**A**) Gene Ontology (GO) classification and enrichment analysis of the differentially expressed genes (DEGs) in shoots between 9311-2x and 9311-4x. (**B**) Kyoto Encyclopedia of Genes and Genomes (KEGG) pathways of the significantly enriched DEGs in shoots between 9311-2x and 9311-4x. (**C**) Heatmap of the expression of genes related to peroxidase superfamily proteins in the shoots of 9311-2x and 9311-4x rice plants. Yellow indicates gene expression levels (read counts per million with log2 value). White indicates the fold change (log2 value) of DEGs. * *p* < 0.05. R2 and R4 represent the root samples derived from 9311-2x and 9311-4x plants, respectively. R2_AS and R4_AS represent the alkaline stress treated root samples derived from 9311-2x and 9311-4x plants, respectively. Likewise, Sh2 and Sh4 represent the shoot samples derived from 9311-2x and 9311-4x plants, respectively. Sh2_AS and Sh4_AS represent the alkaline stress treated shoot samples derived from 9311-2x and 9311-4x plants, respectively.

**Figure 7 ijms-23-05561-f007:**
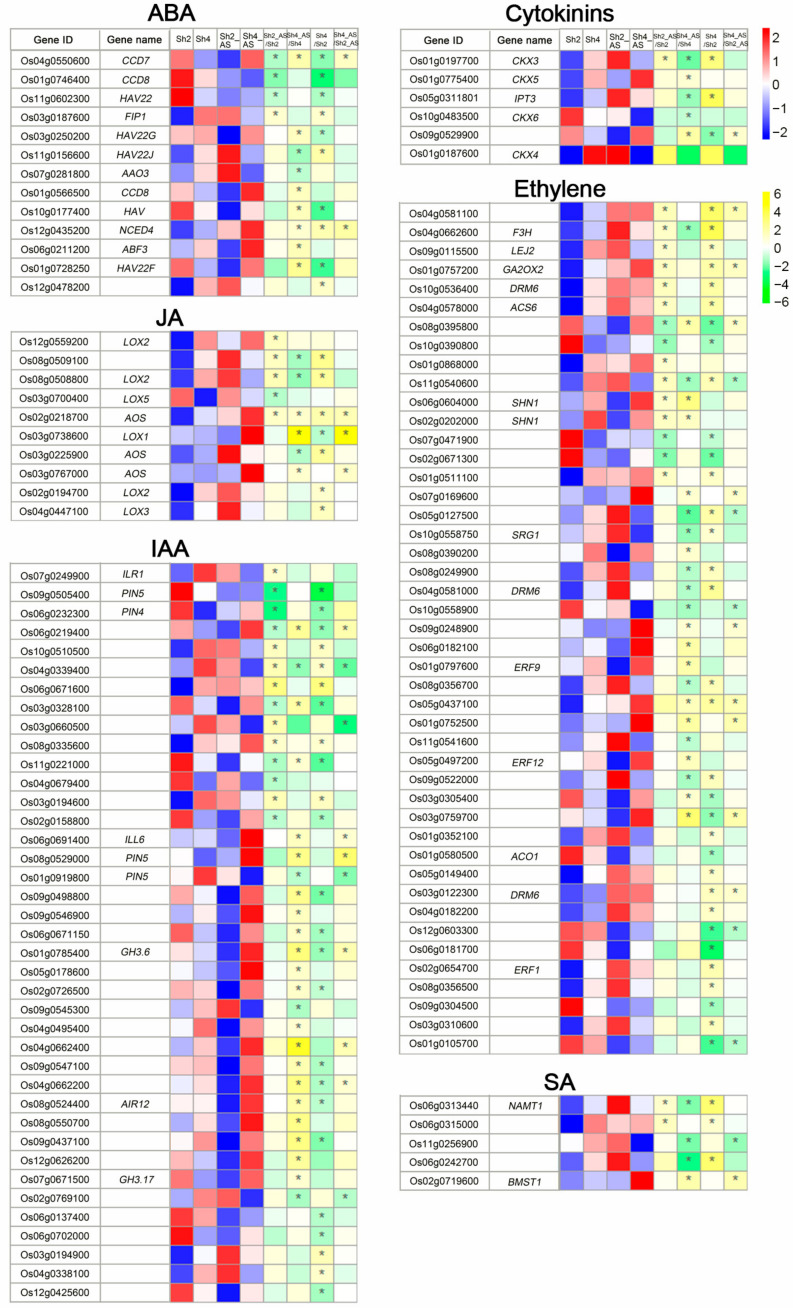
Expression profiles of genes related to phytohormones in shoots between 9311-2x and 9311-4x rice plants under mock and high pH conditions. Heatmap of differentially expressed genes (DEGs) analyzed using MapMan software. Yellow indicates genes expression levels (read_counts per million with log2 value). White indicates the fold change (log2 value) of DEGs. * *p* < 0.05.

## Data Availability

The datasets generated and analyzed in this study are available at PRJNA812638 (https://www.ncbi.nlm.nih.gov/sra/PRJNA812638), accessed on 5 March 2022.

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
