# Peer review of "Alkaline Stress Induces Different Physiological, Hormonal and Gene Expression Responses in Diploid and Autotetraploid Rice"

_ijms, 2022, doi:10.3390/ijms23105561_

Round 1

Reviewer 1 Report

The subject of the study is interesting, it provides a lot of results, and it is high-quality work. The information is properly presented and discussed. In my opinion, it deserves publication after some minor corrections:

The most important thing is to improve the information in the material and methods section. You should describe more the way plants were grown (number of plants per treatment, number of replicates, hydroponic? Pots? Substrate?, % humidity)

In addition, you should describe the methodology of organic material content, enzyme activity, and hormone levels

L502. What do you mean by organic material content? Apparently, you do not include this analysis in this study

L128-129. Figure 1D is not referenced in the figure caption. Dry weight is (C) and fresh weight is (D)?

L199. Delete the extra space between “AS.” and “There”

In figure captions, you write: “Different letters in the same column represent significant difference”. I think is confusing. The different letters indicate significant differences between two separate columns not in the same column. You should change this

In figures 4 and 5 you should specify that “Sh” refers to shoot and “R” to root

L209. Define DEG the first time is mentioned in the text

L260. Define GO and KEGG the first time are mentioned in the text

Reviewer 2 Report

The manuscript of Ningning Wang and co-authors is devoted to the study of alkaline stress responses in diploid and tetraploid rice. The authors provided a large amount of data. However, in my opinion, the question of why tetraploid rice is more resistant to alkaline stress remained open. The authors showed only that diploid and tetraploid plants react differently to alkaline stress. The manuscript should be revised due to the large number of questions, shortcomings and comments.

1) The discussion section contains a number of conclusions that cannot be accepted because for most parameters there is no pattern of only decreasing or increasing in time. But the authors make a general conclusion. In addition, in this section, in most cases, there are no references to figures on the basis of which the authors made one or another conclusion.

Lines 367-370 Significantly the level differs only for MDA 6h (according to Fig. 3). What do the authors think is the reason for this?

Lines 424-427 This statement is true only for 3 and 24 h for ABA and for 3 h for JA (according to Fig. 3)

Lines 438-441 Does not correspond to Fig. 3

Lines 450-453 In which moment (according to Fig. 2)?

Lines 453-455 In which moment and in which plants (according to Fig. 3)? After 24 h for tetraploid rice only. But RNA isolation was performed after 6 h of stress. Therefore, these results cannot be compared.

2) The discussion section needs a number of clarifications

Lines 412-414 What exactly are these hormones?

Lines 461-464 What exactly are these genes? And what was understood from the data obtained?

Lines 469-473 If the data is mentioned here, then the Supp. Fig.6 should be transferred to the text of the article.

3) About pH 11.39 (line 117). Under what conditions such a high pH can occur in nature?

4) Some notes on the Figures

Fig.1 What is it (d)?

Fig. 3 Designations of zeatin in the text and in the figure do not match

Fig. 5A and 6A Text is unreadable

Fig. 6 and 7 After 6 hours?

Fig. 6 It is not clear what is yellow and white

Fig. 6 and 7 The sequence of plant variants in the tables should be the same everywhere to make it easier to compare

5) The methods section is poorly described. There is not even an indication of which method was used in some cases (especially 4.2, but also 4.3 and 4.4).

6) There are no data on qRT-PCR (4.4 Methods section). What are the genes in Supplementary table 2? What do they correspond to? Why is there only one reference gene? Where did the primer structures come from?

7) Supplementary materials do not contain legends of figures and tables.

Round 2

Reviewer 2 Report

The authors have made all necessary changes. The article is suitable for publication.